# Genetically regulated gene expression and proteins revealed discordant effects

**Janne Pott**[1,2]*, **Tarcyane Garcia**[1], **Stefanie M. Hauck**[3], **Agnese Petrera**[3], **Kerstin Wirkner**[1,2], **Markus Loeffler**[1,2], **Holger Kirsten**[1,2], **Annette Peters**[3,4,5], **Markus Scholz**[1,2]*

**1** Institute for Medical Informatics, Statistics and Epidemiology, Medical Faculty, University of Leipzig, Leipzig, Germany, **2** LIFE Research Center for Civilization Diseases, Medical Faculty, University of Leipzig, Leipzig, Germany, **3** Research Unit Protein Science and Metabolomics and Proteomics Core Facility, Helmholtz Zentrum Munich - German Research Center for Environmental Health, Neuherberg, Germany, **4** German Center for Diabetes Research (DZD), München-Neuherberg, Germany, **5** Chair of Epidemiology, Institute for Medical Information Processing, Biometry and Epidemiology, Medical Faculty, Ludwig-Maximilians-Universität München, Munich, Germany

* janne.pott@imise.uni-leipzig.de (JP); markus.scholz@imise.uni-leipzig.de (MS)

## Abstract

### Background

Although gene-expression (GE) and protein levels are typically strongly genetically regulated, their correlation is known to be low. Here we investigate this phenomenon by focusing on the genetic background of this correlation in order to understand the similarities and differences in the genetic regulation of these omics layers.

### Methods and results

We performed locus-wide association studies of 92 protein levels measured in whole blood for 2,014 samples of European ancestry and found that 66 are genetically regulated. Three female- and one male-specific effects were detected. We estimated the genetically regulated GE for all significant genes in 49 GTEx v8 tissues. A total of 7 proteins showed negative correlations with their respective GE across multiple tissues. Finally, we tested for causal links of GE on protein expression via Mendelian Randomization, and confirmed a negative causal effect of GE on protein level for five of these genes in a total of 63 gene-tissue pairs: BLMH, CASP3, CXCL16, IL6R, and SFTPD. For IL6R, we replicated the negative causal effect on coronary-artery disease (CAD), while its GE was positively linked to CAD.

### Conclusion

While total GE and protein levels are only weakly correlated, we found high correlations between their genetically regulated components across multiple tissues. Of note, strong negative causal effects of tissue-specific GE on five protein levels were detected. Causal network analyses revealed that GE effects on CAD risks was in general mediated by protein levels.

**Data Availability Statement:** All summary statistics are publicly available from Zenodo (DOI: 10.5281/zenodo.6045694). Scripts used in the secondary analyses are included at https://github.com/GenStatLeipzig/LWAS_Olink. Complete data

sets including genetic data cannot be shared publicly due to ethical and legal restrictions, as they are sufficient to identity study participants. This is not covered by the informed consent form of the LIFE-Adult study. Data are available from the LIFE Research Center (contact via Dr. Matthias Nüchter, Head of Managing Office, E-mail: matthias. nuechter@life.uni-leipzig.de) for researchers who meet the criteria for access to confidential data.

**Funding:** LIFE-Adult is funded by the Leipzig Research Center for Civilization Diseases (LIFE). LIFE is an organizational unit affiliated to the Medical Faculty of the University of Leipzig. LIFE is funded by means of the European Union, by the European Regional Development Fund (ERDF) and by funds of the Free State of Saxony within the framework of the excellence initiative. Olink measurements were funded by the HI-MAG project "Serum proteome biomarkers as mediators of cardiometabolic disease development" of the Medical Faculty of the University Leipzig and the Helmholtz Zentrum München. The funders had no role in study design, data collection and analysis, decision to publish, or preparation of the manuscript.

**Competing interests:** The authors have declared that no competing interests exist.

## Introduction

Several large-scale genome-wide association studies (GWASs) identified more than 150 genetic risk loci of coronary artery disease (CAD) [1–6]. However, for the majority of loci, the underlying molecular pathology remains to be elucidated. High-throughput proteomics could contribute to our understanding of molecular patho-mechanisms by providing functional causal links between genetic loci, proteome expressions and cardiovascular disease traits.

While there is typically a strong relationship between genetics and transcriptomics via cis expression quantitative trait loci (eQTLs), some studies have shown only weak correlations between the transcriptomic and the proteomic layer [7–10]. Possible reasons comprise different half-lives of mRNA and respective protein, posttranscriptional modifications and tissue and compartment specificity [11]. Nevertheless, genetic effects on expression and protein (proteome quantitative trait loci—pQTL) levels are partly overlapping suggesting common genetic drivers. For example, the Framingham Heart Study detected 26 pQTLs in cis that overlapped with respective eQTLs in whole blood, liver and heart tissues [12]. He et al. [13] analyzed the liver-specific proteome on genome-wide scale and found for about 40% of all tested genes an overlap with known eQTLs in liver.

In this study, we aimed at identifying cis-pQTLs for a panel of 92 biomarkers of CAD measured by proximity extension assays in blood. To characterize these cis loci in more detail, we compared the effects of cis-eQTLs and pQTLs at these loci in more detail. For this purpose, we analyzed the overlap of eQTLs and pQTLs by co-localization analyses and tested for association of genetically regulated gene expression (GE) across tissues and respective blood protein expression (PE). Finally, the identified genetic associations were used to establish causal chains of genetics, transcriptomics, proteomics and CAD via concatenated Mendelian Randomization analyses.

## Material and methods

### Cohort description

All analyses were performed in participants of the LIFE-Adult study. In LIFE-Adult, 10,000 residents of the city of Leipzig, Germany were randomly recruited in an age- and sex-stratified manner. All participants were deeply examined with respect to civilization diseases such as obesity, diabetes, cardiovascular diseases, cognitive impairment and mental disorders as well as contributing environmental and life-style factors. Details can be found in Loeffler et al. [14]. Blood samples were taken from all participants after an overnight fasting and were stored in the Leipzig Medical Biobank for subsequent analyzes and measurements of genetic, transcriptomic and proteomic data. Overlap of OMICs data is displayed in S1 Fig in S1 File.

LIFE-Adult meets the ethical standards of the Declaration of Helsinki and is approved by the Ethics Committee of the Medical Faculty of the University Leipzig, Germany (Reg. No 263-2009-14122009). Written informed consent including agreement with genetic analyses was obtained from all participants. A basic description of samples used in this study can be found in Table 1.

### Protein biomarker measurement

For proteomic profiling, we selected EDTA plasma probes of 2,024 elderly LIFE-Adult participants. Measurement of 92 CVD-related protein biomarkers was performed with the proximity extension assay (PEA) [15] using the Olink CVD Panel III. Measurements were performed in 23 batches each including 88 samples and two identical controls each. For eight samples, Olink

**Table 1. Basic sample description.**

| Variable | Overall | Female | Male | P-value |
|---|---|---|---|---|
| | (n = 2,014) | (n = 974) | (n = 1,040) | |
| Age (years) | 62.5 (11.5) | 62.0 (11.3) | 62.9 (11.6) | **2.47E-02** |
| BMI (kg/m$^2$) | 27.7 (4.5) | 27.2 (4.8) | 28.1 (4.1) | **1.02E-06** |
| Current smoker | 320 (16.7%) | 147 (16.2%) | 173 (17.3%) | 5.49E-01 |
| Hypertension[a] | 1166 (58.7%) | 496 (51.9%) | 670 (65.1%) | **2.91E-09** |
| Type 2 diabetes[a] | 479 (23.8%) | 214 (22.0%) | 265 (25.5%) | 7.25E-02 |
| Statin therapy[b] | 319 (15.9%) | 124 (12.8%) | 195 (18.8%) | **2.99E-04** |
| TC (mmol/l) | 5.70 (1.06) | 5.87 (1.06) | 5.53 (1.04) | **3.35E-13** |
| LDL-C (mmol/l) | 3.58 (0.95) | 3.59 (0.96) | 3.57 (0.95) | 5.88E-01 |
| HDL-C (mmol/l) | 1.62 (0.46) | 1.81 (0.46) | 1.44 (0.39) | **2.86E-73** |

For continuous parameters, the unit is given in parenthesis, and arithmetic mean and standard deviation values are shown. For binary variables, total number and percentages are provided. Differences between sexes were tested with a chi-squared test for all binary parameters, and with Mann-Whitney U test for all continuous parameters. Abbreviations: BMI, body mass index; TC, total cholesterol; LDL-C, low-density lipoprotein; HDL-C, high-density lipoprotein.

[a] anamnestic, medication or determined by HbA1c>6.5%;

[b] ATC-code beginning with C10.

measurement failed, and additional two samples were excluded as outliers (Mahalanobis Distance >3 IQR), resulting in N = 2,014 samples available for further analyses.

Measurements of biomarkers are available for all of these samples except for three biomarkers. Two assays (BLMH, CTSD) failed at one plate, resulting in N = 1,926 for these traits. A single missing value of metalloproteinase 4 was mean-imputed. Across all 92 assays in CVD III, the mean intra-assay (within run) and inter-assay (between runs) variations expressed as coefficients of variation are reported to be 8.1% and 11.5%, respectively. We used normalized protein expression units as semi-quantitative traits. Genetic data were available for all of the samples. An overview of all biomarkers including their distribution and genetic regions is given in S1 Table in S2 File.

### Gene expression measurement

Isolated mRNA from whole blood of 3,527 samples was hybridized to Illumina HT-12 v4 Expression BeadChips (Illumina, San Diego, CA, USA) and gene expression (GE) was measured on the Illumina HiScan (47,231 raw GE probes). We then processed the data by log2-transformation, quantile-normalization [16, 17] and correction for batch effects [18] using R/Bioconductor.

Probes were excluded if they were (1) expressed in less than 5% of the samples, (2) still significantly associated with batch effects, or (3) unable to map to a gene according to ingenuity pathway analyses (IPA, QIAGEN Inc., accessed on 2019-04-04). In summary, 20,972 valid GE probes remained, corresponding to 15,950 genes. We looked for transcripts corresponding to the 92 proteome features of the PEA. There were 91 probes with sufficient QC, matching to 68 unique genes.

Samples were removed if (1) the number of detected GE probes deviated more than 3*IQR from the median, (2) the Mahalanobis distance of several quality characteristics deviated more than 3*IQRs from the median [19], or (3) the Euclidean distance of expression values deviated more than 4*IQRs from the median [16]. Overall, of the assayed 3,527 samples, 110 had to be

removed for quality reasons. Of those remaining, 3,194 samples had also genetic data available in high quality.

## Genotyping & Imputation

A total of 7,838 participants of LIFE-Adult were genotyped on the genome-wide SNP array Axiom CEU1 (Affymetrix). Genotype calling was performed using the software Affymetrix Power Tools (version 1.20.06). We conducted calling and quality control according to Affymetrix's best practice steps [20].

SNPs were excluded if (1) their call rate was less than 97%, (2) there was a significant violation of Hardy-Weinberg equilibrium ($p < 1x10^{-6}$ for autosomal SNPs, $p < 1x10^{-4}$ for X-chromosomal SNPs in women only), (3) significant plate association ($p < 1x10^{-7}$), or (4) cluster plot specific parameters according to Affymetrix's recommendation [20].

Samples were removed if (1) their signal contrast on the array was low ($< 0.82$), (2) their call rate was less than 97%, (3) the estimated sex differed from the sex retrieved from the databank, (4) cryptic relatedness was observed ($> 0.6$ [21]), or (5) the estimated genetic ethnicity was out of range ($> 6^{*}SD$ in any of the first 10 principal components). There were 33 ethnic outliers, which were removed for all further analyses (see S2 Fig in S1 File). After filtering, LIFE-Adult was genetically homogeneous and we therefore refrained from correcting for population stratification via PCs in the main analyses, but included the first ten PCs in a sensitivity analysis of all lead SNPs per protein.

We imputed our SNP data on the reference 1000 Genome Phase 3 [22] using SHAPEIT [23] v2r900 for prephasing and IMPUTE2 [24] v2.3.2 for genotype estimation. For this study, all SNPs with minor allele frequency (MAF) $<1\%$ or imputation info score $<0.8$ were excluded, resulting in 9,033,656 SNPs for further analyses.

## Statistical analysis

An overview of our analysis plan is given in S3 Fig in S1 File.

**Genetic association analyses for 92 protein biomarkers.** For each of the 92 biomarkers we performed genetic association analyses at the regions of the gene coding for the biomarker, i.e. we searched for cis-pQTLs, only. The region between gene start -500 kb and gene stop +500 kb was considered (see S1 Table in S2 File for the assumed starts and stops of genes). Primary genetic association analysis was done in all subjects (n = 2,014) with PE adjusted for age and sex. In a secondary analyses, we ran sex-stratified analyses (n = 974 female, n = 1,040 male) adjusting PE for age. For the analyses, we used the additive frequentist model with expected genotype counts as implemented in PLINK 2.0 [25]. We lifted our data from hg19 to hg38 using the GWAS Summary Statistics harmonization tool [26].

We pooled all cis-regions of our primary analysis and performed a hierarchical FDR correction as suggested for eQTLs by Peterson et al. [27]. In more detail, we first applied Benjamini & Hochberg (BH) [28] correction of all SNPs associations calculate for a specific PE and identified the SNP with the minimal corrected p-value (Simes p-value). Next, we applied BH on the 92 Simes p-values and tested with $\alpha_1 = 0.05$ to determine the $k$ proteins showing significant associations. We then used $\alpha_2 = 0.05 \times k/92$ as significance threshold on the first level as proposed by Benjamini & Bogomolov (BB) [29]. The SNP with lowest and significant p-value was denoted as lead cis-pQTL of the respective protein. We then merged all significant associations and pruned the variants to a subset of markers that are in approximate linkage equilibrium with each other ($r^2 < 0.1$). Linkage disequilibrium (LD) was calculated using all LIFE-Adult participants. Finally, we annotated these independent variants with (1) other nearby genes (Ensemble, +/- 250 kb of SNP position) [30], (2) known traits associations from the GWAS

Catalog (LD $r^2>0.3$) [31], (3) known cis-eQTLs (LD $r^2>0.3$, $\alpha_{cis} = 0.05$) [32–35], and (4) CADD scores as measure of deleteriousness [36]. We defined novel loci as regions whose lead SNP was not in LD with a variant reported for blood protein biomarker levels in the GWAS catalog (LD $r^2 \leq 0.3$).

For all lead cis-pQTLs, we checked for sex-specific effects on PE and compared effect sizes between females and males applying t-tests of beta estimates [37]. We also looked for sex-specific significant loci by applying the same hierarchical FDR correction as mentioned above. Finally, we looked up the eQTL summary statistics of all GTEx v8 [38] tissues for all lead SNPs and their associated genes and compared their effect direction with our pQTL findings. The GTEx data used for the analyses described in this manuscript were obtained from the GTEx Portal on 09.06.2020. We reported those for whole blood and the (second) best associated tissue. In addition, per locus we retrieved the best cis-eQTLs (defined by lowest p-value per tissue and gene) and calculated pairwise LD ($r^2$) to the respective lead pQTLs. To validate our findings, we performed whole blood cis-eQTL analysis in our LIFE data (using n = 3,194 samples with gene expression in whole blood and genetic data).

**Co-localization and association analyses between gene-expression and protein levels.** In order to investigate the link between gene expression and protein levels in more detail, we performed three locus-wise analyses: First, we performed a pairwise co-localization test [39] between our pQTLs and eQTLs obtained from GTEx v8. In more detail, this method tested if two trait associations share the same causal variant, regardless of effect direction. Five hypotheses were tested in parallel, of which H4 states that the traits share the causal SNP, while H3 assumes two independent signals. As threshold for co-localization and for independence, we used a posterior probability for H4 and H3 of $\geq 0.75$, respectively. The region of co-localization was defined as the position of the lead pQTL +/- 500 kb. We used the R-package "coloc" for this analysis [39].

In a second analysis, the summary statistics of all proteomic features with significant cis-pQTLs were used to search for correlations with respective genetically estimated gene-expression (gGE) using the MetaXcan approach [40]. The expression prediction models were downloaded from the github repository [41] (see also PredictDB [42]; GTEx v8 models using elastic net algorithm). PredictDB contains only models that passed stringent criteria (e.g., number of SNPs used, posterior probability for being an eQTL). Hence, not all gene—tissue combinations were available for this analysis. In total, we tested 2,242 tissue-specific gGE for protein association. To adjust for multiple testing of several tissues per protein, we performed a hierarchical FDR correction as mentioned above. The first level were the tissues per protein, the second level were the analyzed proteins. We report findings in whole blood and the (second) best associated tissue.

Finally, we validated the MetaXcan results obtained for blood tissue using our measured gene-expression profiles. Raw gene-expression data were available for 45 of the 64 genes with pQTLs and paired GE/proteome data were available for 1048 samples. We estimated both Pearson's correlation and Pearson´s partial correlation controlling for sex, age, percentage of lymphocytes and percentage of monocytes on total white blood cells. We repeated this analysis in the sex-stratified subsets.

**Mendelian randomization analyses.** To investigate whether observed associations between GE and PE were causal, we performed Mendelian Randomization (MR) analyses. As MR requires strong instruments, we used the best-associated cis-eQTLs per tissue with lowest p-value and p$<5x10-8$ (n = 428 SNPs for a total of 58 genes). To adjust for multiple testing, we performed a hierarchical FDR correction as mentioned above. The first level were the tissues, the second level were the analyzed genes.

Since the proteins on our array were supposed to be cardio-vascular biomarkers, we also estimated the causal effects of protein levels on CAD. We considered only lead pQTLs reaching $p < 5 \times 10^{-8}$ as instruments (n = 48 proteins). P-values of MR were adjusted using Bonferroni correction for 48 tests. For proteins with significant causal effect on CAD, we tested for causal chains GE → PE → CAD [43].

In all analyses, we used the ratio method and estimated the standard error using the first two terms of the delta method [44]. Summary statistics were obtained from our pQTL-analyses, from GTEx v8 [38] and from van der Harst et al. [6].

## Results

An overview of all 92 analyzed proteins, their abbreviations and full name is given in S1 Table in S2 File. In the following, only the gene name abbreviations are used, with the regular written names referring to PE and italic written names to GE. In addition, all main results are included in this table as TRUE/FALSE vectors, which summarize the genetic associations, the GE correlations and the causal analyzes per protein (for the combined setting).

### SNP level results

After applying hierarchical FDR, we detected for 64 biomarkers significant associations in or nearby the corresponding gene (23,951 unique SNPs, see S2 Table in S2 File for an overview of all Simes p-values). Priority pruning revealed 758 independent SNPs (see S3 Table in S2 File for summary statistics and full annotation with nearby genes, GWAS catalog traits and enriched pathways, and S4 Table in S2 File for lead pQTLs per protein). Several of these loci were already described for association with blood protein levels (n = 27 loci) [45–49]. Of the remaining 37 loci, 25 were previously reported for other traits (e.g. lipids, CAD related traits, or blood fractions), while 12 loci were not reported for any trait associations so far. These 37 loci are considered novel for our protein traits. Fig 1 shows a circular plot of all cis regions and $-\log_{10}$ transformed association p-values of our association study and those of the best GTEx tissue per gene. Although we did not perform a classic GWAS, 16 of the 37 novel and 22 of the other loci also reached the classic genome-wide significance threshold of $p < 5 \times 10^{-8}$ (regarding the Simes p-value).

In our sex-stratified approach, we detected 54 proteins significantly associated in men, and 48 in women (see S2 Table in S2 File). Five proteins were associated in males, but not in females, and had lower Simes p-values compared to the combined setting, suggesting male-specific loci (GDF15, MPO, PAI, OPN, and TFF3). Of note, PAI was only associated in the male setting, not in the combined one. Similarly, NOTCH3 was only associated in the female setting, but not in the combined or male setting. In the following, all 66 proteins with association in at least one setting are analyzed. Regarding the 110 lead SNPs of all settings (both from the combined and sex-stratified analyses, if other lead SNPs were detected here), we observed for 18 of them a significant difference in effect size in men and women, but only five of them survived multiple testing correction (S4 Fig in S1 File), including NOTCH3 and MPO. We reported the sex-stratified results in S4 Table in S2 File. In our sensitivity analyses additionally adjusting for the first ten principal components, we found no significant bias in our results, as the effect estimates were the same, and their p-value increased only slightly. All associations remained significant according to our FDR threshold (see S4 Table in S2 File).

Next, we searched the GTEx database for associations of our 110 lead pQTLs with GE of the corresponding genes. We found 7,714 such associations across all 49 GTEx tissues. We checked the direction consistency of the e- and pQTLs, and surprisingly, detected for 41 proteins at least one discordant direction. For 13 of them, this discordant direction was observed

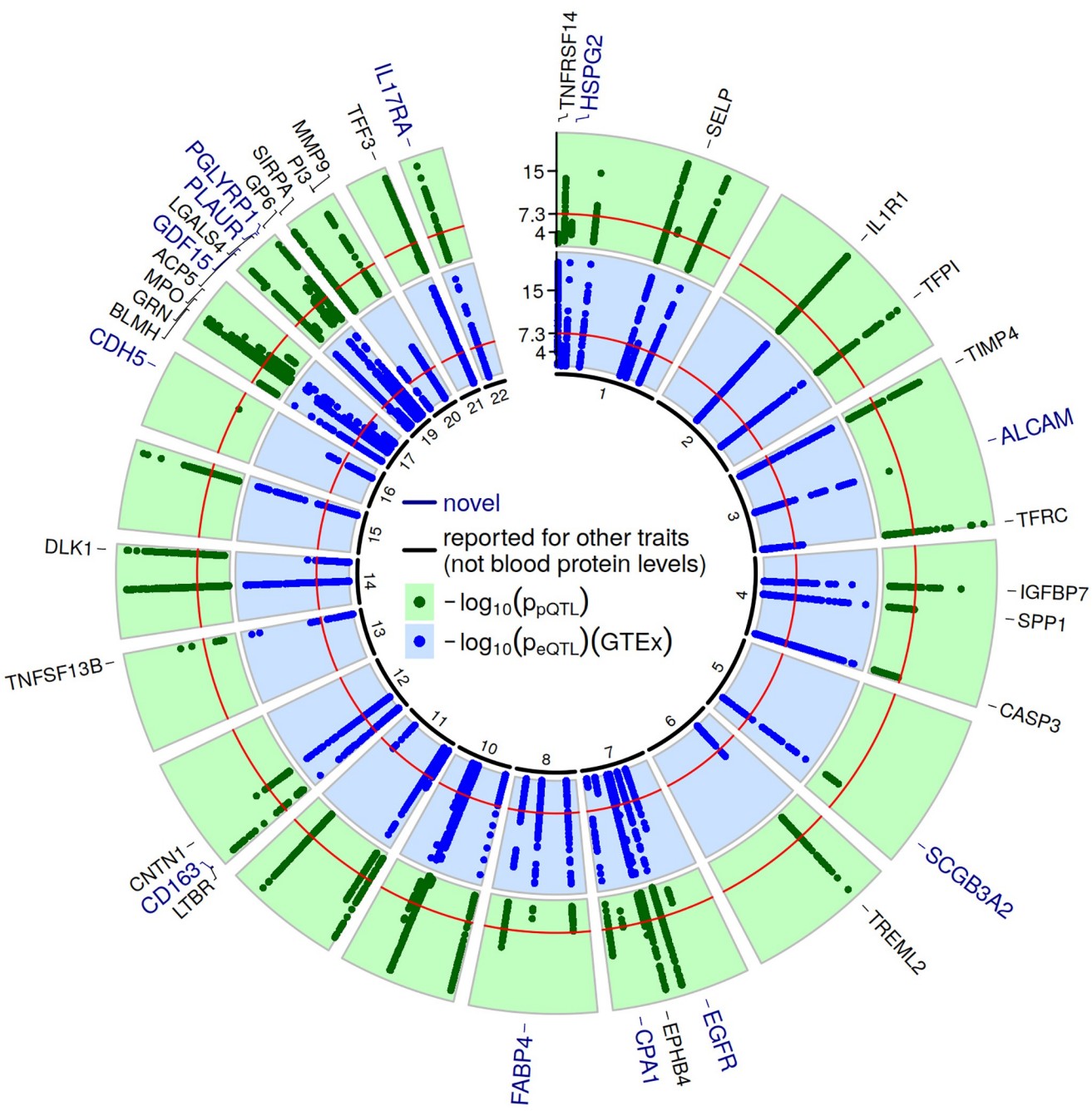

**Fig 1. Circular plot of cis-associations.** Log-transformed p-values for cis-pQTLs and eQTLs are shown in the green respectively blue circle. We obtained the statistics for eQTLs from GTEx and present the results of the tissue with the strongest eQTL per gene (see S3 Table in S2 File). For plotting, the y-axis was restricted to -log (p) = 20, i.e. all larger -log (p) values were set to 20. The red circles mark the classical genome-wide significance threshold (p = 5x10[-8]). Gene names are added for loci not yet described for blood protein levels, and are colored with respect to the novelty level (blue: not described for any other traits, black: reported for other traits except for blood protein levels).

in most of the associated tissues (more than 75% of tissues in which the eQTL was observed, see Table 2 and Fig 2). Restricted to whole blood, there were ten QTLs with discordant effects. To validate this finding in whole blood, we replicated the eQTL analysis in our LIFE data (GE available for 45 genes). Here, four of the ten SNPs were associated with p<0.05 and showed

**Table 2. Comparison of effect direction of cis-pQTLs from our GWAS and cis-eQTLs from GTEx.**

| Locus Information | | pQTL | | eQTL GTEx whole blood and (sec.) best tissue | | |
|---|---|---|---|---|---|---|
| Protein (ratio) | pQTL effect allele / EAF | beta | p-value | beta | p-value | Tissue |
| IL6R | rs4129267 | 0.421 | $2.96 \times 10^{-323}$ | -0.087 | $5.40 \times 10^{-09}$ | Artery Tib. |
| (17/18) | T / 0.380 | | | -0.194 | $8.41 \times 10^{-17}$ | |
| CCL15 | rs41436444 | 0.939 | $4.56 \times 10^{-292}$ | - | - | Lung |
| (20/21) | CAGGGCAG / 0.080 | | | -0.635 | $1.65 \times 10^{-17}$ | |
| CCL16 | rs10445391 | -0.852 | $8.01 \times 10^{-155}$ | - | - | Thyroid |
| (3/4) | G / 0.072 | | | 0.220 | $2.27 \times 10^{-04}$ | |
| **SFTPD** | rs721917 | -0.475 | $1.37 \times 10^{-95}$ | 0.113 | $3.39 \times 10^{-03}$ | Artery Tib. |
| (42/42) | G / 0.406 | | | 0.661 | $1.90 \times 10^{-43}$ | |
| **BLMH** | rs7214248 | 0.193 | $4.29 \times 10^{-61}$ | -0.072 | $3.80 \times 10^{-04}$ | Artery Tib. |
| (24/26) | A / 0.346 | | | -0.221 | $1.02 \times 10^{-17}$ | |
| **ACP5** | rs897811 | -0.163 | $1.70 \times 10^{-20}$ | 0.501 | $5.78 \times 10^{-19}$ | Thyroid |
| (35/37) | C / 0.116 | | | 0.376 | $9.71 \times 10^{-10}$ | |
| TIMP4 | rs392394 | 0.140 | $4.54 \times 10^{-17}$ | -0.024 | $4.36 \times 10^{-01}$ | Artery Tib. |
| (16/17) | A / 0.782 | | | -0.239 | $7.95 \times 10^{-10}$ | |
| TNFRSF11B | rs11300005 | 0.073 | $2.70 \times 10^{-11}$ | - | - | Eso. Mus. |
| (2/2) | C / 0.493 | | | -0.133 | $1.32 \times 10^{-02}$ | |
| AXL | rs3786556 | -0.446 | $9.39 \times 10^{-16}$ | -0.002 | $9.68 \times 10^{-01}$ | Artery Tib. |
| (28/28) | T / 0.184 | | | 0.227 | $2.13 \times 10^{-13}$ | |
| CPA1 | rs35454128 | 0.197 | $3.79 \times 10^{-08}$ | 0.117 | $8.18 \times 10^{-02}$ | Adipose Sub. |
| (7/9) | C / 0.120 | | | -0.237 | $1.95 \times 10^{-03}$ | |
| CASP3 | rs6845294 | 0.153 | $1.09 \times 10^{-07}$ | 0.026 | $1.54 \times 10^{-01}$ | Cells fibro. |
| (14/15) | A / 0.687 | | | -0.248 | $5.35 \times 10^{-17}$ | |
| CDH5 | rs16956504 | 0.105 | $1.10 \times 10^{-07}$ | 0.007 | $8.93 \times 10^{-01}$ | Pituitary |
| (3/3) | C / 0.110 | | | -0.161 | $2.14 \times 10^{-02}$ | |
| **CXCL16** | rs145042193 | -0.064 | $8.22 \times 10^{-07}$ | 0.104 | $2.47 \times 10^{-03}$ | Cells fibro. |
| (25/25) | T / 0.208 | | | 0.362 | $4.92 \times 10^{-17}$ | |

We show results of the 13 genes for which discordant effect directions between pQTL and most of the respective eQTLs (more than 75% of all significant eQTLs across tissues) were observed. We also report eQTLs of whole blood and the best-associated tissue in GTEx. The effect allele and its frequency is given below of the respective SNP ID. For four genes we could replicate the different effect direction in our LIFE data (marked in bold, see S2 Table in S2 File for more details).

discordant effect directions when compared to the corresponding pQTL (SFTPD, BLMH, ACP5, and CXCL16). The other SNPs had the same effect direction or showed no significant effect in our data.

## Locus level results

Although all lead pQTLs were associated with GE in at least one tissue, only in 4% of the GE-tissue combinations the best eQTL was also the best pQTL, and in 30% the two SNPs were in some LD ($r^2 > 0.1$). To determine whether these signals are inter-related, we performed co-localization analyses and tested for an association of genetically regulated gene expression and protein levels. A summary of these tissue-specific analyses is shown as Venn diagram (S5 Fig in S1 File) and in S6 Table in S2 File.

We observed 50 proteins with at least one shared (PP4>75%) GE signal, and 42 with at least one independent signal (PP3>75%) across tissues. A total of 34 proteins show both, dependent and independent signals in different tissues. Posterior probabilities for all pairs can

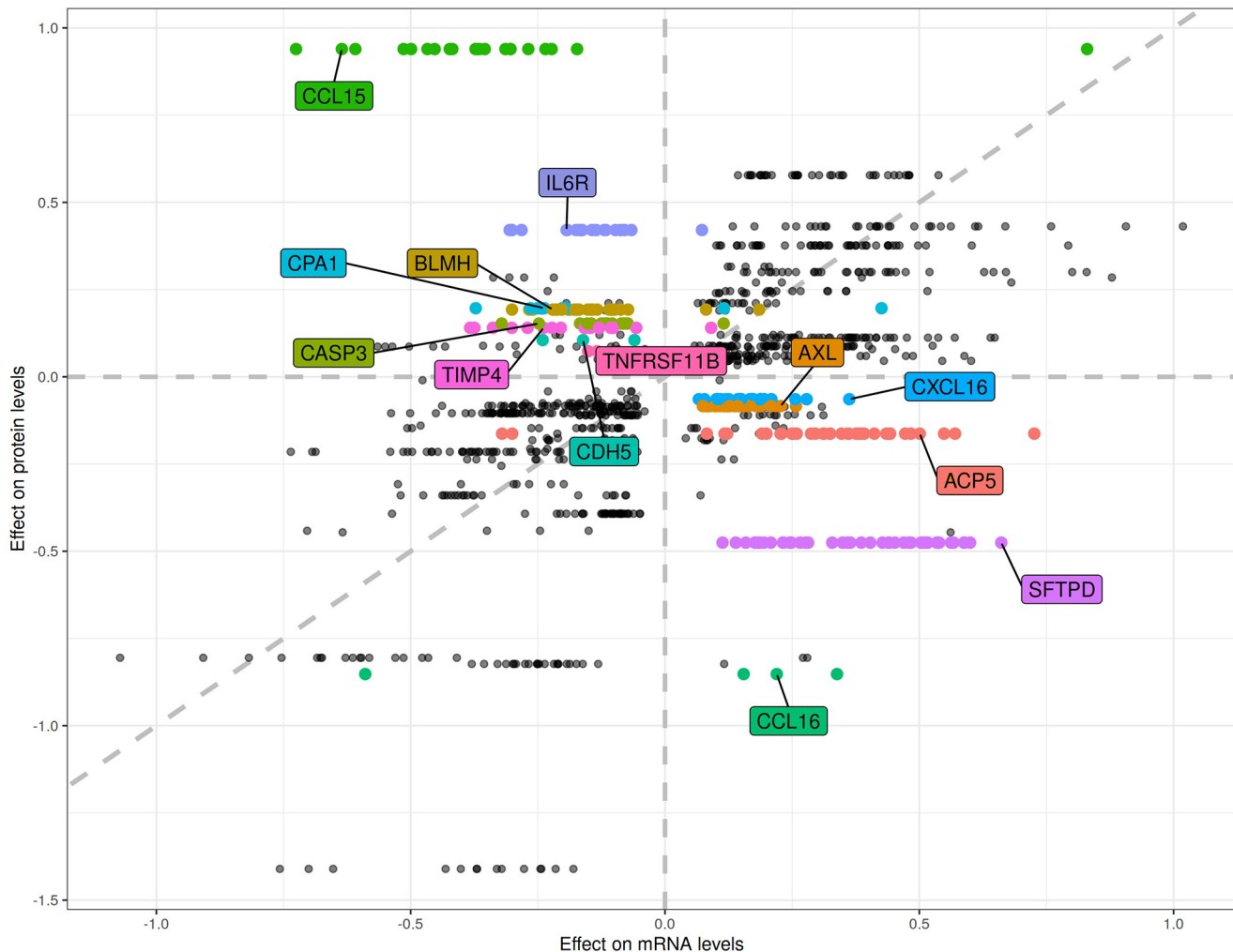

**Fig 2. Scatter plot of effect estimates of the 63 lead pQTL-SNPs on gene expression and protein levels.** As the focus was on the direction only, we did not normalize the effect estimates. Only SNPs with a significant eQTL in at least one tissue are displayed (p< = 0.05 in GTEx). Results of the 13 genes showing discordant pQTL and eQTL directions in more than 75% of eQTL tissues are labeled (see also Table 2 and S5 Table in S2 File).

be found in S7 Table in S2 File and are displayed in S6 Fig in S1 File. We compared the gene-tissue combinations of shared and independent signaling with those of high and low LD between best eQTL and pQTL. Regarding the higher LD ($r^2 > 0.1$) combinations, the distribution between shared and independent signals was almost the same (n = 212 with PP3>0.75, n = 295 with PP4>0.75). This demonstrates that LD does not guarantee co-localization. In contrast, for low LD pairs, there was a clear trend to independent signals (n = 189 with PP3>0.75, n = 14 with PP4>0.75).

We estimated the genetically regulated gene expression in all GTEx v8 tissues using MetaX-can. Here, several SNPs at each gene locus were selected and included into the GE prediction model. The predicted GE was then tested for association with the respective protein. After applying hierarchical FDR, we detected significant associations for 58 of the 64 considered biomarkers in at least one tissue (n = 1,474 significant tests of a total of 2,242). Most genes showed this association in about half of all tissues (median of 27 associated tissues). Counterexamples are LBTR and IL17RA, showing GE-PE association across many tissues (49, respectively 42 tissues).

**Table 3. Proteins showing predominantly negative correlation between tissue-specific GE and blood PE.**

| Protein | #tissues (neg/tot) | Effect | P-value | Tissue |
|---|---|---|---|---|
| IL6R | 25/29 | -2.177 | $7.11 \times 10^{-303}$ | Artery Tib. |
| | | -2.965 | $8.21 \times 10^{-197}$ | WB |
| SFTPD | 33/40 | -6.003 | $7.69 \times 10^{-77}$ | Colon Trans. |
| BLMH | 12/15 | -0.985 | $1.48 \times 10^{-58}$ | Artery Tib. |
| | | -1.052 | $8.11 \times 10^{-15}$ | WB |
| TIMP4 | 29/36 | -0.371 | $8.00 \times 10^{-17}$ | Brain Ant. Cingulate Cortex |
| CASP3 | 25/29 | 0.865 | $1.51 \times 10^{-7}$ | Brain Putamen |
| | | -109.229 | $2.78 \times 10^{-7}$ | WB |
| CXCL16 | 41/41 | -0.242 | $6.70 \times 10^{-7}$ | Kidney Cortex |
| | | -0.067 | $1.54 \times 10^{-2}$ | WB |
| TREML2 | 13/14 | -40.610 | $5.85 \times 10^{-6}$ | Thyroid |
| | | -0.876 | $3.58 \times 10^{-4}$ | WB |

MetaXcan results are shown for the best-associated tissue with negative effect estimate and whole blood (WB), if a prediction model was available. Neg = Number of significant negative correlations across tissues. Tot = Total number of significant associations.

We compare the MetaXcan results with our results obtained by co-localization analyses. The intersection of significant associations and co-localization comprised n = 531 gene-tissue pairs, of which n = 288 pairs showed co-localization and n = 243 indicated independent signaling. This demonstrates that results of MetaXcan-based gene-expression association analysis are only loosely related to those of co-localization. We summarized all MetaXcan results in S8 Table in S2 File.

We checked the direction of the correlation of tissue-specific GE and PE. A total of 42 proteins showed opposite direction of effects in at least one tissue. Seven of them showed this negative association in most of the tissues (see Table 3), including six which were also found based on the LD considerations performed in the previous paragraph (see Table 2).

Finally, we compared the MetaXcan-derived GE-protein associations with GE-protein associations based on raw GE data of whole blood from our LIFE study. GE data were available for 45 genes. Among those, we detected 21 significant partial correlations controlling for age, sex and white blood cell counts (S9 Table in S2 File). Comparing these results with the respective MetaXcan results of whole blood, we found eight pairs that are significant in both analyses. For all of them, the same effect direction was observed, with negative correlation of PE, and both, total GE and genetically estimated GE in whole blood.

## Causal network of gene expression, protein levels and CAD

We performed Mendelian Randomization analyses of the causal relationship between GE and PE for all tissues for which a strong eQTL ($p<5\times10^{-8}$) was available. Accordingly, we tested 58 genes in up to 27 tissues (n = 670 tests in the combined setting). We detected causal links for 51 genes (501 genes-tissue pairs). There were predominantly positive effects (364 pairs with positive causal effect of GE on PE). A summary of all instruments, tissues, and causal effects is given in S10 Table in S2 File.

For 419 pairs of GE and protein, we found both, significant MetaXcan association and MR effect. Among those, 398 showed concordant effect directions between GE and PE (92 pairs with negative effect, 306 with positive effect), i.e. effect directions are in large agreement. The 92 pairs with negative effect comprise 20 unique genes. Twelve of them show this relation in >75% of associated tissues, including five genes described in the previous sections (BLMH,

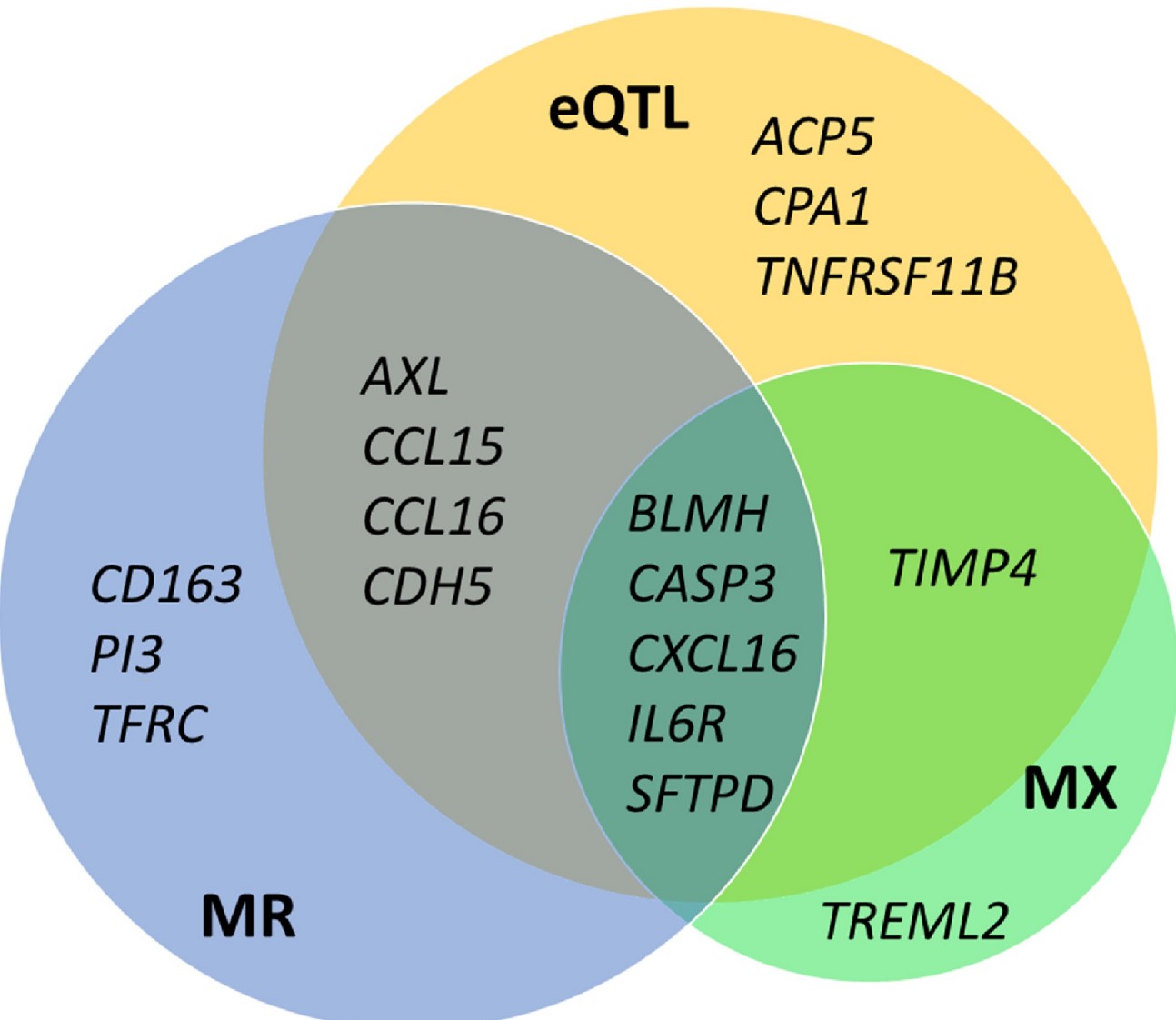

**Fig 3. Overlap of genes with negative correlation to protein levels according to different analysis strategies.** We present the genes with negative protein correlations for more than 75% of the significant tissues according to MetaXcan (MX), negative causal effect estimates according to Mendelian Randomization (MR), and opposite effect direction of eQTLs and pQTLs (see also Tables 2–4).

CASP3, CXCL16, IL6R, and SFTPD). Summaries of analysis results of these genes are shown in Fig 3 and S7 Fig in S1 File, and Table 4.

Next, we tested the causal link between the biomarkers and coronary artery disease. We restricted the analyses to lead pQTLs with $p < 5 \times 10^{-8}$ and available CAD statistics from van der Haarst et al. [6]. This left us with 47 biomarkers. Results are given in S11 Table in S2 File and a scatter plot for all pairs is shown in S8 Fig in S1 File. Four protein showed a significant effect in the combined setting: IL6R ($\beta_{IV} = -0.094$, $p = 4.90 \times 10^{-14}$), PCSK9 ($\beta_{IV} = 0.540$, $p = 4.72 \times 10^{-10}$), TFPI ($\beta_{IV} = -0.116$, $p = 7.83 \times 10^{-5}$), and AXL ($\beta_{IV} = 0.487$, $p = 6.09 \times 10^{-5}$). IL6R had significant causal estimates in the sex-stratified settings as well. The estimated causal TFPI effect was also significant in females, but in males it reached only nominal significance that did not

**Table 4. Proteins with predominantly negative causal links of GE and PE.**

| Protein | #tissues (neg/tot) | Causal effect estimate | P-value MR | Association effect estimate | P-value MetaXcan | Tissue |
|---|---|---|---|---|---|---|
| IL6R | 5/5 | -0.713 | $5.57 \times 10^{-27}$ | 0.126 | $4.63 \times 10^{-06}$ | Testis |
| | | -3.008 | $1.35 \times 10^{-16}$ | -2.965 | $8.21 \times 10^{-197}$ | WB |
| SFTPD | 22/25 | -0.347 | $2.64 \times 10^{-17}$ | -0.050 | $5.68 \times 10^{-04}$ | Heart AA |
| | | -0.749 | $6.43 \times 10^{-06}$ | | | WB |
| BLMH | 2/2 | -0.873 | $3.07 \times 10^{-15}$ | -0.985 | $1.48 \times 10^{-58}$ | Artery Tib. |
| CCL15 | 15/16 | -0.459 | $1.02 \times 10^{-14}$ | | | Nerve Tib. |
| PI3 | 1/1 | -0.731 | $1.02 \times 10^{-08}$ | | | WB |
| AXL | 6/6 | -0.243 | $6.03 \times 10^{-06}$ | | | Artery Tib. |
| CCL16 | 15/15 | -0.181 | $1.28 \times 10^{-05}$ | | | Liver |
| CASP3 | 5/6 | -0.440 | $1.89 \times 10^{-05}$ | -0.173 | $1.57 \times 10^{-06}$ | Cells fibro. |
| | | 0.914 | $1.83 \times 10^{-05}$ | 0.865 | $1.51 \times 10^{-07}$ | WB |
| TFRC | 1/1 | -0.404 | $7.94 \times 10^{-05}$ | | | Lung |
| CXCL16 | 17/17 | -0.225 | $1.69 \times 10^{-04}$ | -0.213 | $1.68 \times 10^{-05}$ | Thyroid |
| | | -0.097 | $2.97 \times 10^{-03}$ | -0.067 | $1.54 \times 10^{-02}$ | WB |
| CDH5 | 1/1 | -0.148 | $1.92 \times 10^{-02}$ | | | Thyroid |
| CD163 | 1/1 | -0.129 | $3.66 \times 10^{-02}$ | | | Testis |

MR results are shown for the best tissue with negative effect estimate and whole blood, if available. We added respective MetaXcan results for comparison.

Neg = Number of significant negative causal estimate across tissue. Tot = Total number of significant MR tests.

survive multiple testing correction. PCSK9 was also causally linked in males, while in females the PCSK9 instrument did not reach the significance threshold to be included for MR analysis. For AXL, both sex-stratified instruments were above the MR significance threshold and hence excluded.

Finally, we searched for causal chains from GE over PE to CAD for all four proteins with significant causal link to CAD, which were also causally affected by gene expression (n = 50 tests). The total causal effect estimates of GE on CAD were significant in 26 gene-tissue pairs and negative in four of them (*TFPI* in tibial artery and cultured fibroblast cells in all and females). *AXL* was not causally linked to CAD in any tissue. The indirect effect was estimated as product of the GE → PE and PE → CAD effects, which corresponds to the effect of GE on CAD mediated by PE. These indirect effect estimates were significant for all 50 gene-tissue pairs and for 45 of them no significant difference between the total and indirect GE-CAD effect was observed, indicating complete mediation of GE via PE towards CAD. We summarized the results in S12 Table in S2 File and displayed the causal chains in Fig 4.

## Discussion

In this work, we performed a genetic cis-association analysis of 92 cardiovascular biomarkers and found 66 regulations. We used these signals to unravel the relationship of cis-pQTLs and cis-eQTLs by (1) testing for co-localization of signals, (2) analyzing the correlation of genetically regulated GE and PE, and (3) testing for causal effects of the GE/PE associations. Finally, we established causal chains of GE, PE and CAD across tissues.

In our study, we focus on cis-effects rather than a whole-genome hypothesis-free approach. Similar to eQTL analyses, cis-effects tend to be true positives, while trans-effects are often false positives requiring much more stringent false positive control limiting power of this type of analysis. We check the GWAS catalog for known associations, and detected that 37 of our

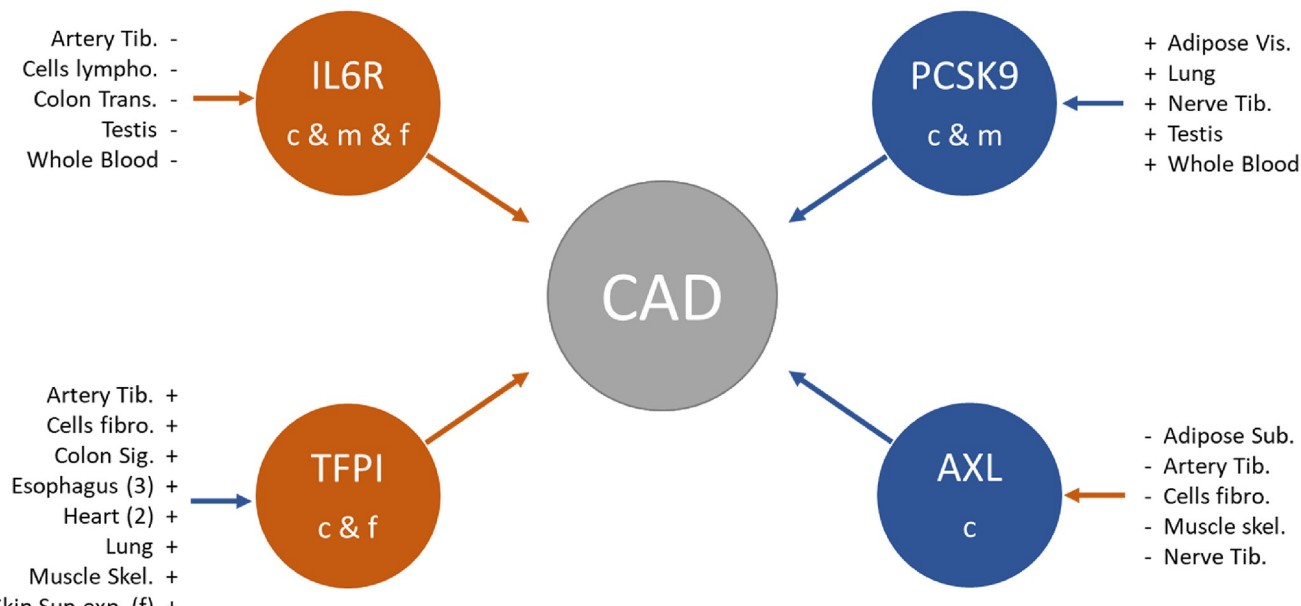

**Fig 4. Graphical overview of causal networks form gene expression (GE) over protein expression (PE) on the outcome coronary artery disease (CAD).** Orange arrows indicate a negative causal effect of protein level on CAD (IL6R and TFPI) or of gene expression on protein levels, blue arrows denote positive links. The settings are indicated by c (combined), m (males) and f (females). Tissues in which gene expression showed a significant indirect effect on CAD are listed next to their gene. Bold tissues indicate also a significant direct effect.

associations are novel. Moreover, for 38 of the 66 associated proteins, our lead SNP achieved genome-wide significance ($p < 5 \times 10^{-8}$).

As example, PI3 was a genome-wide significant and novel loci, which was also associated in the sex-stratified analyses. Here, it showed a significant sex-related effect, i.e. the effect estimate in men was twice that of women. We tested our own GE and eQTL data for this interaction but could not confirm this sex-dimorphism at the GE layer. Moreover, we detected a co-localization of eQTLs and pQTLs for men (whole blood), but not for women. In both MetaXcan and MR, the estimates were significant in each sex, but with stronger effect in men (MR in whole blood: $\beta_{men}$ = -0.990, $\beta_{women}$ = -0.481, $p_{IA}$ = 0.002). *PI3* codes for elafin, which has been linked to the inflammatory response in atherosclerosis [50] and myocardial infarction [51]. Most atherosclerotic outcomes show sexual dimorphism as well, with higher risk for men. This makes elafin an interesting target for further studies of sex-dimorphisms in cardiovascular research.

To unravel the relationship between GE and PE, we performed several analyses across different tissues since the origin of plasma PE is not necessarily whole blood. We found that the lead pQTL was in most cases not the best-associated eQTL. We tested pairwise LD between eQTLs and pQTLs and found most of them in low LD ($r^2 < 0.1$) across tissues and proteins. When comparing LD results with our co-localization results, we found as expected that amongst the low LD pairs signals are often predicted as independent. More surprisingly, for the high LD pairs, the ratio between independence and co-localization was balanced, i.e. LD does not ensure co-localization of the signals. For example, both IL17RA and SFTPD have high LD pairs in 34 and 39 tissues, respectively. IL17RA showed co-localization of these signals in all these tissues. In contrast, for SFTPD, co-localization was refuted for all these signals.

In general, we observed a good agreement of MetaXcan and MR results although the MetaXcan approach does not show causality *per se* and is also based on different gene-models

compared to the instruments used for MR. In contrast, we observed only a moderate overlap between co-localization results and MetaXcan / MR results. Signals with co-localization but no significant causal estimate could be explained by pleiotropic effects while causality but lack of co-localization could be explained by locus heterogeneity, i.e. different causal e- and pQTLs. However, most of the detected associations were found in for high LD pairs of e- and pQTL (Co-localization: 295 of 313 gene tissue pairs [94.2%]; MetaXcan: 659 of 1,474 [44.7%], MR: 357 of 501 [71.3%]).

Most interestingly, we detected five gene-protein pairs, which consistently showed opposite effect directions of eQTLs and pQTLs, negative correlation of GE and PE (MetaXcan) and negative causal effects (Mendelian Randomization). Those were BLMH, CASP3, CXCL16, IL6R and SFTPD. While BLMH was co-localized in 10 of 12 analyzed tissues, the other four had one co-localizing tissue each, but all other tissues suggested independent signals. Only BLMH was co-localized in most tissues. We discuss the functional relevance of this observation in more detail by a GTEx v8 look-up of gene expression levels by tissue [38]. While the highest rate of BLMH expression occurs in skin tissue, it is also expressed in artery tissues, for which we observed negative association and in tibial artery tissue also negative causal effect. High expression levels in GTEx v8 [38] and negative links in our analyses were also found for CXCL16 in tissues testis, whole blood, and skin, and for IL6R in tissues muscle skeletal, whole blood, esophagus muscularis, and colon transverse. CASP3 had negative links between GE and PE in several tissues, but also four positive links for brain (substantia nigra), liver, pancreas and whole blood tissue. The negative links were detected in tissues with higher GE, e.g. cells cultured fibroblasts. SFTPD is highly expressed in lung tissue, in which we found independent e- and pQTL signals and a positive causal estimates. In other tissues with lower SFTPD expression, negative links were observed, e.g. thyroid. Thus, in the last case, the relationship of GE and PE could be dominated by single tissues showing a positive correlation, while for the other four the negative links happen in tissues with substantial gene expression. This suggests functional relevance of BLMH, CASP3, CXCL16, and IL6R that needs further biological validation.

Consistent negative effects between GE and PE are of particular interest to further studies, since the mechanism behind this is not clear. Explanations for this observation could be (1) tissue-specific protein levels that differ from those measured in whole blood; (2) whole blood acting only as transport compartment to a specific target tissue; (3) upregulation of pathways, in which the protein is further metabolized; (4) post-translational modifications that influence protein degradation; or (5) upregulation of genes in response to increased consumption / degradation of protein.

Among our results, IL6R showed the most pronounced effects, with causal negative effects in whole blood, testis, artery tibial and colon transverse tissues. We speculate that inflammatory conditions consume IL6R resulting in low plasma levels but increased gene-expression to counter the IL6R loss. The negative effect of IL6R on CAD was previously reported by Yuan et al. [52] and could be explained by reduced inflammation. However, the GE effect on CAD was positive, which is mainly mediated by IL6R PE.

In our MR mediation analyses, we found several indirect causal links of tissue-specific GE over PE in whole blood on CAD. Only for TFPI, we found indirect effects from heart-specific tissues (atrial appendage and left ventricle). Although these two tissues might be more specific for CAD, cis-effects are usually shared across tissues, with exception of brain tissues [38]. Hence, other tissues such as muscle and whole blood with larger sample sizes might still detect the true contribution of the gene expression. In addition, we do not know which tissue our measured proteins come from. Therefore, it is also possible that increased GE in CAD-unrelated tissues lead to higher blood protein levels. The proteins can be transported to heart tissues, where they increase the risk for an event. The detected direct effects of *AXL* in four

tissues could be false positives, given the comparison to a non-significant total effect. The direct effect of PCSK9 gene expression occurs in Adipose tissue (visceral omentum), where it is only weakly expressed (TPM = 0.27). This needs further biological validation.

One limitation of this study was its relatively small sample size, and with it, reduced power. We therefore refrained from analyzing trans-pQTLs and focused on cis-effects. Larger studies of meta-analyses are required to resolve this limitation. The observed trans-associations could then be used as independent instruments for a bivariate Mendelian Randomization analysis, checking if high protein levels showed reverse causality on the gene expressions, an issue which could not be addressed by our study.

In conclusion, we discovered several causal links of tissue-specific gene-expression and blood protein levels of cardiovascular biomarkers. Observed negative causal relationships are of interest for further studies to unravel the underlying post-transcriptional or pathway-associated regulatory processes. Finally, we established a causal patho-mechanistic network of GE and PE of IL6R, PCSK9, TFPI, and AXL and coronary artery disease providing possible new therapy targets.

## Supporting information

**S1 File.**
(PDF)

**S2 File.**
(XLSX)

## Acknowledgments

We thank all study participants of the LIFE-Adult study whose personal dedication and commitment have made this project possible. LIFE-Adult genotyping (round 3) was done at the Cologne Center for Genomics (CCG, University of Cologne, Peter Nürnberg and Mohammad R. Toliat). For genotype imputation, compute infrastructure provided by ScaDS (Dresden/Leipzig Competence Center for Scalable Data Services and Solutions) at the Leipzig University Computing Centre was used.

## Author Contributions

**Conceptualization:** Markus Scholz.

**Data curation:** Tarcyane Garcia, Stefanie M. Hauck, Agnese Petrera, Kerstin Wirkner, Holger Kirsten.

**Formal analysis:** Janne Pott.

**Funding acquisition:** Markus Loeffler, Annette Peters.

**Investigation:** Janne Pott, Stefanie M. Hauck, Agnese Petrera, Markus Loeffler, Markus Scholz.

**Methodology:** Janne Pott.

**Project administration:** Markus Scholz.

**Resources:** Markus Loeffler, Annette Peters.

**Software:** Holger Kirsten.

**Supervision:** Markus Scholz.

**Visualization:** Janne Pott.

**Writing – original draft:** Janne Pott.

**Writing – review & editing:** Markus Scholz.

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
