## [Decision Letter · Decision Letter 0]

26 Oct 2021

PONE-D-21-20087Genetically regulated gene expression and proteins revealed discordant effectsPLOS ONE

Dear Dr. Pott,

Thank you for submitting your manuscript to PLOS ONE. After careful consideration, we feel that it has merit but does not fully meet PLOS ONE’s publication criteria as it currently stands. Therefore, we invite you to submit a revised version of the manuscript that addresses the points raised during the review process. You may notice the reviewer raised some concerns regarding the methodology. I agree with these points, especially it would be useful to know whether the use of GTEx V8 can replicate the findings based on GTEx V7, as well as the basis of outcome selection.

We look forward to receiving your revised manuscript.

Kind regards,

Jie V Zhao

Academic Editor

PLOS ONE

Reviewers' comments:

Reviewer's Responses to Questions

**Comments to the Author**

1. Is the manuscript technically sound, and do the data support the conclusions?

Reviewer #1: Yes

2. Has the statistical analysis been performed appropriately and rigorously? 

Reviewer #1: Yes

3. Have the authors made all data underlying the findings in their manuscript fully available?

Reviewer #1: Yes

4. Is the manuscript presented in an intelligible fashion and written in standard English?

Reviewer #1: Yes

5. Review Comments to the Author

Reviewer #1: The authors did an extensive work to study the 92 proteins in the whole blood from 2014 participants in both SNP level and locus level. Of which, 63 proteins were genetically regulated, and 3 sex specific effects were identified. The authors then tested for the colocalization between pQTLs and eQTLs and found evidence of both independent and shared signals. Additionally, the authors tested for the association of genetically regulated gene expression (GE) across tissues and identified 10 proteins of negative correlations. Finally, the authors tested the causal link of genomics, transcriptome, proteome to coronary artery disease (CAD) risk and confirmed a negative causal effect of GE on protein level for 5 genes. The main finding was the gene IL6R.

Comment 1: In the statistical analysis, the population stratification (PS) is not considered. It would be great if the authors can include more details to justify why PS is not a concern in the study. Otherwise, if there exists any PS, not adjusting for it could induce significant bias.

Comment 2: GTEx v8 data has been released for quite a while. I wonder why the authors used v7 instead of v8?

Comment 3: The proteins in the study were supposed to be cardiovascular (CVD) biomarkers and the authors tried to establish the link from genomics, transcriptome, proteome to CAD risk. I wonder why the authors included CVD-unrelated tissues, especially when testing for the causal path involving CAD, how does the GE and PE in colon, lung and testis contribute to CAD outcome?

Comment 4: The authors may need to review and correct the typos and formatting issues throughout. For example, 1) in the abstract, CAD is first mentioned and needs to be defined; 2) The table 2 needs to be re-formatted and the minor allele of CCL15 does not look right. In line 238, the results show ten genes instead of eight; 3) The gene names should be in italics format throughout; 4) The figure 2-4 in the main text are mis-numbered as figure 1-3.

6. PLOS authors have the option to publish the peer review history of their article (what does this mean?). If published, this will include your full peer review and any attached files.

Reviewer #1: No

---

## [Author Response · Author response to Decision Letter 0]

11 Feb 2022

Editorial Comments:

1. PLOS ONE's style requirements, including those for file naming

Author reply: We thank the editor for his comments and changed the manuscript accordingly.

2. Minimal Data Set / 3. Data Availability

Author reply: Complete data sets including genetic data cannot be made publicly available due to ethical and legal restrictions, as they are sufficient to identity study participants. This is not covered by the informed consent. However, access to the LIFE-Adult data is possible via project agreements addressed to:

Leipziger Forschungszentrum für Zivilisationserkrankungen (LIFE) 

Dr. Matthias Nüchter

Head of Managing Office

Universität Leipzig, Medizinische Fakultät

Philipp-Rosenthal-Str. 27 

04103 Leipzig 

E-mail: matthias.nuechter@life.uni-leipzig.de

The pQTL summary statistics are publicly available at Zenodo (DOI: 10.5281/zenodo.6045694). If necessary, all R-scripts used for this publication can be made available at github. 

4. Direct billing option

We confirm that the corresponding authors, Janne Pott & Markus Scholz, are affiliated with the University of Leipzig, which is the chosen institute.

Reviewer #1: 

The authors did an extensive work to study the 92 proteins in the whole blood from 2014 participants in both SNP level and locus level. Of which, 63 proteins were genetically regulated, and 3 sex specific effects were identified. The authors then tested for the colocalization between pQTLs and eQTLs and found evidence of both independent and shared signals. Additionally, the authors tested for the association of genetically regulated gene expression (GE) across tissues and identified 10 proteins of negative correlations. Finally, the authors tested the causal link of genomics, transcriptome, proteome to coronary artery disease (CAD) risk and confirmed a negative causal effect of GE on protein level for 5 genes. The main finding was the gene IL6R.

Comment 1: In the statistical analysis, the population stratification (PS) is not considered. It would be great if the authors can include more details to justify why PS is not a concern in the study. Otherwise, if there exists any PS, not adjusting for it could induce significant bias.

Author reply: The LIFE Adult study is a long-term, population-based cohort study with participants from Leipzig and surrounding areas. In line with this, the genetic homogeneity is high (see new Supplemental Figure 3). We therefore decided to remove the few ethnic outliers and refrained from adjustment for principal components. However, we repeated our association tests for SNPs after hierarchical FDR and adjusted for 10 principle components. The beta estimates were essentially the same, while the standard error increased slightly. All SNPs remained significant according to our hierarchical FDR threshold (see Supplemental Table 4). 

Changes in manuscript: We added a PCA plot of our study data as new Supplemental Figure 3 revealing high homogeneity, and included PC adjusted statistics for the lead SNPs in Supplemental Table 4. We also described our approach in the Material section in more detail.

Comment 2: GTEx v8 data has been released for quite a while. I wonder why the authors used v7 instead of v8?

Author reply: We thank the reviewer very much for this suggestion. Previously, we used hg19 SNP data, which is the same build as in GTEx v7. We agree that the most recent version of GTEx should be used and have therefore harmonized our data to hg38 and repeated all our analysis with the harmonized data and GTEx v8. 

Changes in manuscript: We updated the methods for the harmonization and GTEx v8 usage. Please note that this has changed all of the tables and figures, while the main message of the paper remains the same.

Comment 3: The proteins in the study were supposed to be cardiovascular (CVD) biomarkers and the authors tried to establish the link from genomics, transcriptome, proteome to CAD risk. I wonder why the authors included CVD-unrelated tissues, especially when testing for the causal path involving CAD, how does the GE and PE in colon, lung and testis contribute to CAD outcome?

Author reply: We agree to discuss this in more detail. Cis-effects have been reported to be shared across tissues, with exception of brain tissues [The GTEx Consortium, DOI: 10.1126/science.aaz1776]. Hence we used also different tissues, partly with larger sample size, which might still detect the true contribution of the gene expression on CAD. Our described links of GE to CAD are mediated by PE, suggesting that increased GE in CAD-unrelated tissues lead to higher blood protein levels, which are then transported to heart tissues, where they increase the risk for a cardiovascular event. 

Changes in manuscript: We added a paragraph in the Discussion section.

Comment 4: The authors may need to review and correct the typos and formatting issues throughout. For example, 1) in the abstract, CAD is first mentioned and needs to be defined; 2) The table 2 needs to be re-formatted and the minor allele of CCL15 does not look right. In line 238, the results show ten genes instead of eight; 3) The gene names should be in italics format throughout; 4) The figure 2-4 in the main text are mis-numbered as figure 1-3.

Author reply: We are sorry for these typos. 

Changes in manuscript: We checked our manuscript after the extensive changes. We believe we have now eliminated all typos and formatting issues.

---

## [Decision Letter · Decision Letter 1]

26 Apr 2022

PONE-D-21-20087R1Genetically regulated gene expression and proteins revealed discordant effectsPLOS ONE

Dear Dr. Pott,

Thank you for submitting your manuscript to PLOS ONE. After careful consideration, we feel that it has merit but does not fully meet PLOS ONE’s publication criteria as it currently stands. Therefore, we invite you to submit a revised version of the manuscript that addresses the points raised during the review process. As you can see, one reviewer has pointed out some wording issues and consistency throughout the paper, please revise accordingly.

We look forward to receiving your revised manuscript.

Kind regards,

Jie V Zhao

Section Editor

PLOS ONE

Journal Requirements:

Reviewers' comments:

Reviewer's Responses to Questions

**Comments to the Author**

1. If the authors have adequately addressed your comments raised in a previous round of review and you feel that this manuscript is now acceptable for publication, you may indicate that here to bypass the “Comments to the Author” section, enter your conflict of interest statement in the “Confidential to Editor” section, and submit your "Accept" recommendation.

Reviewer #1: All comments have been addressed

Reviewer #2: All comments have been addressed

2. Is the manuscript technically sound, and do the data support the conclusions?

Reviewer #1: Yes

Reviewer #2: Yes

3. Has the statistical analysis been performed appropriately and rigorously? 

Reviewer #1: Yes

Reviewer #2: Yes

4. Have the authors made all data underlying the findings in their manuscript fully available?

Reviewer #1: No

Reviewer #2: No

5. Is the manuscript presented in an intelligible fashion and written in standard English?

Reviewer #1: Yes

Reviewer #2: Yes

6. Review Comments to the Author

Reviewer #1: The authors did extensive work to revise the manuscript. Now the manuscript looks great. I do not have further question.

Reviewer #2: In the revised version of their manuscript the authors have addressed the comments raised by the reviewer and doing so the quality and clarity of the manuscript have improved substantially.

I have only a couple of (very) minor further comments:

1. Figure numbers in the main text are still not correct.

2. The authors may want to present numbers in a consistent way throughout the manuscript, either as e.g. "2384" or "2,384".

3. It seems at several occasions there is a typo with r22 instead of r2 (when referring to LD), and line 313 “data WERE available”.

4. The authors may want to check for consistent use of past tense in the material and methods section, e.g. line 135 and line 196 in the revised version of the manuscript.

5. Data from the 1000 Genome Project Phase 3 have been used for genotype imputation. I wonder, why this rather old dataset was used instead of a more updated sources for imputation, e.g. the Haplotype reference consortium r1.1 .

6. I find the term "best eQTL" a bit unclear and would like the authors to specify whether they refer to strongest/most significant eQTL and/or largest effect size.

7. A more cautious wording with regard to Mendelian Randomization may be more adequate, as this methods gives an estimate of causation/causality but does not prove or guarantee such relationship, e.g. line 349 "IL6R was the only protein that showed causality in all three settings".

8. In the part of the discussion section where the authors refer to TPM values of several genes in several tissues, many new results are reported (that otherwise have only appeared in supplementary material). I suggest to limit the content of new data presented here and instead report these findings in the results section.

7. PLOS authors have the option to publish the peer review history of their article (what does this mean?). If published, this will include your full peer review and any attached files.

Reviewer #1: No

Reviewer #2: No

---

## [Author Response · Author response to Decision Letter 1]

6 May 2022

Reviewer #1: 

The authors did extensive work to revise the manuscript. Now the manuscript looks great. I do not have further question.

Authors reply: We thank the reviewer very much for the positive evaluation. 

Reviewer #2: 

In the revised version of their manuscript the authors have addressed the comments raised by the reviewer and doing so the quality and clarity of the manuscript have improved substantially.

I have only a couple of (very) minor further comments:

Authors reply: We thank the reviewer very much for the encouraging evaluation and the helpful comments.

Comment 1: Figure numbers in the main text are still not correct.

Author reply: We are sorry for not checking the created PDF file for this mix-up. For Fig 1, the corresponding text box was accidently deleted. Hence, when converting to a PDF, the auto-numbering used 1 for Fig 2, and so on. 

Changes in manuscript: We added the text box for Fig 1 again. 

Comment 2: The authors may want to present numbers in a consistent way throughout the manuscript, either as e.g. "2384" or "2,384".

Author reply: We are sorry for this mix-up and present numbers now consistently as “2,014”. 

Changes in manuscript: We changed all number so that the thousands separator is consistently a comma. 

Comment 3: It seems at several occasions there is a typo with r22 instead of r2 (when referring to LD), and line 313 “data WERE available”.

Author reply: We believe this to be a marked-up problem, as we did not find this typo in the cleaned manuscript. We accidently deleted the superscripted 2, and added it again. However, in the marked-up mode, a superscripted deleted number looks like an underlined number and hence it reads as r22 instead of r2. 

Changes in manuscript: We checked the manuscript for the r22 typo and corrected line 313. 

Comment 4: The authors may want to check for consistent use of past tense in the material and methods section, e.g. line 135 and line 196 in the revised version of the manuscript.

Author reply: We are sorry for these inconsistencies and corrected them in the manuscript

Changes in manuscript: We checked the Material & Methods section and corrected the grammar if necessary. 

Comment 5: Data from the 1000 Genome Project Phase 3 have been used for genotype imputation. I wonder, why this rather old dataset was used instead of a more updated sources for imputation, e.g. the Haplotype reference consortium r1.1 .

Author reply: We were not able to legally update imputation of genetic data of the LIFE consortium to newer references such as HRC or TopMed, as these references are not publicly accessible for imputation on our servers. Therefore, genetic data of the LIFE-consortium would have to be transferred to e.g. Michigan Imputation Server in the United States of America (US) or Sanger Imputation Server in the United Kingdom (UK). For both US and UK, there are still open Data Privacy concerns, as the level of data protection does not comply with the standards required by the European Union (EU). For the US, the Trans-Atlantic Data Privacy Framework is under development, but not yet in place (see https://ec.europa.eu/commission/presscorner/detail/en/ip_22_2087). For the UK, an adequacy decision under the General Data Protection Regulation had been adopted by the EU just last June. However, the final decision for imputation on a server in a country other than an EU member state lies within the LIFE consortia. We will update imputation of the LIFE genetic data once the consortium agrees, or an alternative Imputation Server is available within the EU. 

Changes in manuscript: None.

Comment 6: I find the term "best eQTL" a bit unclear and would like the authors to specify whether they refer to strongest/most significant eQTL and/or largest effect size.

Author reply: We meant the cis-eQTL with lowest p-value per gene and tissue. 

Changes in manuscript: We added this information in the Material and Method section. 

Comment 7: A more cautious wording with regard to Mendelian Randomization may be more adequate, as this methods gives an estimate of causation/causality but does not prove or guarantee such relationship, e.g. line 349 "IL6R was the only protein that showed causality in all three settings".

Author reply: We thank the reviewer for this reminder and agree to tone down the statements on causality. 

Changes in manuscript: We used a more cautious wording in the Results section regarding Mendelian Randomization. In addition we added some information why not all settings were tested for this analysis. 

Comment 8: In the part of the discussion section where the authors refer to TPM values of several genes in several tissues, many new results are reported (that otherwise have only appeared in supplementary material). I suggest to limit the content of new data presented here and instead report these findings in the results section.

Author reply: In this part of the discussion, we tried to evalute the functional relevance by checking the published expression rate of GTEx v8 of the identified genes by tissues, e.g. do these negative links only occur in tissues with low expressions. Hence, that are not new results. However, we agree to shorten that paragraph by focusing on the main comparisions only. 

Changes in manuscript: We shortend the TPM paragraph in the discussion and clarified its reference, GTEx v8.

---

## [Editor Report · Decision Letter 2]

10 May 2022

Genetically regulated gene expression and proteins revealed discordant effects

PONE-D-21-20087R2

Dear Dr. Pott,

We’re pleased to inform you that your manuscript has been judged scientifically suitable for publication and will be formally accepted for publication once it meets all outstanding technical requirements.

Kind regards,

Jie V Zhao

Section Editor

PLOS ONE
---

## [Editor Report · Acceptance letter]

12 May 2022

PONE-D-21-20087R2 

Genetically regulated gene expression and proteins revealed discordant effects 

Dear Dr. Pott:

I'm pleased to inform you that your manuscript has been deemed suitable for publication in PLOS ONE. Congratulations! Your manuscript is now with our production department. 

Kind regards, 

on behalf of

Dr. Jie V Zhao 

Section Editor

PLOS ONE